# Influenced Zone of Deep Excavation on Adjacent Tunnel Displacement and Control Effect of Ground Improvement in Soft Soil

Bo Liu [1,*], Chengmeng Shao [1,2] and Wen Xu [3]

1 School of Civil Engineering, Southeast University, Nanjing 211189, China
2 China Railway 16th Bureau Group Co., Ltd., Beijing 100018, China
3 College of Civil Aviation, Nanjing University of Aeronautics and Astronautics, Nanjing 211106, China
* Correspondence: boliu@seu.edu.cn

**Abstract:** The aim of this study is to predict the influenced zone of deep excavation on adjacent tunnel displacement, evaluate the control effect of ground improvement, and give the optimal parameters for ground improvement. Based on the current research, a series of finite element method (FEM) numerical simulations were conducted to study the deep excavation-induced tunnel displacement behaviors, considering different tunnel positions outside the pit. On this basis, the influenced zone of deep excavation on an adjacent tunnel was divided corresponding to 3-level tunnel displacement control standards. Then, the commonly used control measure of ground improvement was chosen to study the effects of strength, depth, and width of the improved soil outside the pit on the displacement behaviors of the tunnel. An index of tunnel displacement control effectiveness ($\eta$) was proposed to quantitively characterize the control effect on tunnel displacement. Considering the control effect and engineering economy, the suggested values of strength, depth, and width of the improved soil were provided. Finally, the control effect of ground improvement outside the pit on the influenced zone of deep excavation was studied using the suggested parameters. The research indicates that the range outside the pit can be divided into: I—primary influenced zone, II—secondary influenced zone, III—general influenced zone, and IV—weak influenced zone. Considering the control effect and engineering economy, it is suggested that the ground improvement strength should be kept within 1.5~2 MPa, the ground improvement depth should be 2 times the excavation depth, and the ground improvement width should be increased as much as possible if the site condition allows. After the ground improvement using the suggested parameters, the scope of the influenced zone of deep excavation is reduced and the I—primary influenced zone no longer exists.

**Keywords:** deep excavation; existing tunnel; influenced zone; ground improvement; control effect

## 1. Introduction

Currently, with the rapid development of underground space and urban rail transit, the phenomenon of deep excavation adjacent to existing tunnels is increasing [1–4]. Excavation can inevitably disturb the surrounding rock and soil, and then cause the displacement of the existing tunnel buried in it [5–8]. This disturbed area is called the influenced zone of deep excavation. Therefore, it is important to predict the influenced zone of deep excavation and take measures to reduce the tunnel displacement in the influenced zone before excavation.

At present, the simplest method to determine the influenced zone is using laws and regulations. For example, the management regulations of urban rail transit in Beijing, Shanghai, Nanjing, etc., take 50 m beyond the outer line of the subway tunnel as the protection zone for tunnel deformation. In this method, a fixed value is used to determine the scope of the influenced zone of construction, which is convenient for management departments to control risks. However, this method does not consider the effects of geological conditions, tunnel structure forms, construction workmanship, etc.

In the code GB 50911–2013 [9], according to the different influence degrees, the range outside the deep excavation is divided into: I—primary influenced zone, II—secondary influenced zone, and III—possible influenced zone. This method is simple and easy to understand, but the parameter for dividing the influenced zone is only the excavation depth $H$, without considering the other factors such as geological conditions, deformation forms and magnitudes of the retaining structures. Additionally, the intersection line of the pit wall and base slab is taken as the baseline for dividing the influenced zone, which is more suitable for the traditional shallow excavation using the retaining structure of a flexible sheet pile or without a retaining structure. For the deep and large excavations with deeply embedded retaining structures, the rationality of this simplification is debatable.

Ding (2009) [10] put forward another division method of the influenced zone of deep excavation considering the excavation-induced soil deformation behaviors. The range outside the pit is divided into: I—primary influenced zone, II—secondary influenced zone, and III—uninfluenced zone. Different from the triangular influenced zone in the GB 50911-2013 [9], the influenced zone in this method is simplified as a right-angled trapezoid, and the bottom of the retaining structure is taken as the baseline for dividing the influenced zone. However, the scope of the influenced zone has not been quantitively characterized.

Zheng et al. (2016, 2017) [11,12] studied the displacement behaviors of existing tunnels induced by adjacent deep excavation and, on this basis, the range outside the pit is divided into: I—primary influenced zone, II—secondary influenced zone, III—general influenced zone, and IV—weak influenced zone. In his method, the relationship between the displacement control standards of the tunnel and the influenced zone of deep excavation is established, and the displacement of the tunnel can be directly predicted according to the actual influenced zone where it locates.

Referencing the method proposed by Zheng et al. (2016, 2017) [11,12], Fan et al. (2021) [13] studied the excavation-induced displacement behaviors of underlying tunnels and then divided the range below the base slab into: I—primary influenced zone, II—secondary influenced zone, III—general influenced zone, and IV—weak influenced zone. On this basis, Pu and Liu (2021) [14] studied the effects of ground improvement below the base slab on the underlying tunnel displacement and the influenced zone of deep excavation.

The above studies played an important role in determining the influenced zone of deep excavation on tunnel displacement. However, the study on the control of the influenced zone and tunnel displacement is still limited. In this study, numerical simulations were conducted to analyze the tunnel displacement behaviors induced by adjacent deep excavation considering the different tunnel positions outside the pit, and on this basis, the influenced zone of deep excavation on tunnel displacement was divided corresponding to 3-level tunnel displacement control standards. Then, the commonly used control measure of ground improvement was chosen to study the effects of strength, depth, and width of the improved soil outside the pit on the displacement behaviors of the tunnel. An index of displacement control effectiveness ($\eta$) was proposed to quantitively characterize the control effect on tunnel displacement. Considering the control effect and engineering economy, the suggested values of strength, depth, and width of the improved soil were provided. Finally, the control effect of ground improvement outside the pit on the influenced zone of deep excavation was evaluated using the suggested parameters.

## 2. Influenced Zone of Deep Excavation on Adjacent Tunnel

### 2.1. Numerical Simulation of the Effect of Deep Excavation on Adjacent Tunnel

2.1.1. FEM Numerical Model

Studying tunnel displacement behaviors is the basis for dividing the influenced zone of deep excavation on adjacent tunnels. Here, a series of numerical simulations were conducted using finite element software MIDAS GTS NX. Figure 1 shows the 2D FEM numerical model used to study the effect of deep excavation on adjacent tunnel displacement. The supporting structure of the deep excavation adopts a diaphragm wall (abbreviation DW) and horizontal struts. Considering that the excavation depth of the underground

2-story subway station and the underground 3 to 4-story building basement is about 18 m, the excavation depth ($H_e$) is set at 18 m, the half-width of excavation is set at 30 m, the retaining wall thickness is set at 0.8 m, and the insertion depth below the base slab ($H_i$) is set equal to the excavation depth ($H_e$). The first level of horizontal struts is 1 m below the ground surface, and the vertical interval between the other levels of struts is 4.5 m. The existing tunnel outside the pit is a typical subway shield tunnel with a diameter of 6.0 m and a wall thickness of 0.3 m, the horizontal distance from the tunnel center to the excavation boundary is $L_t$, and the buried depth of the tunnel center is $H_t$, which are analyzed as variables. The length of the model outside the deep excavation is 120 m, and the depth below the base slab is 54 m, which can ensure that the boundary effect is eliminated.

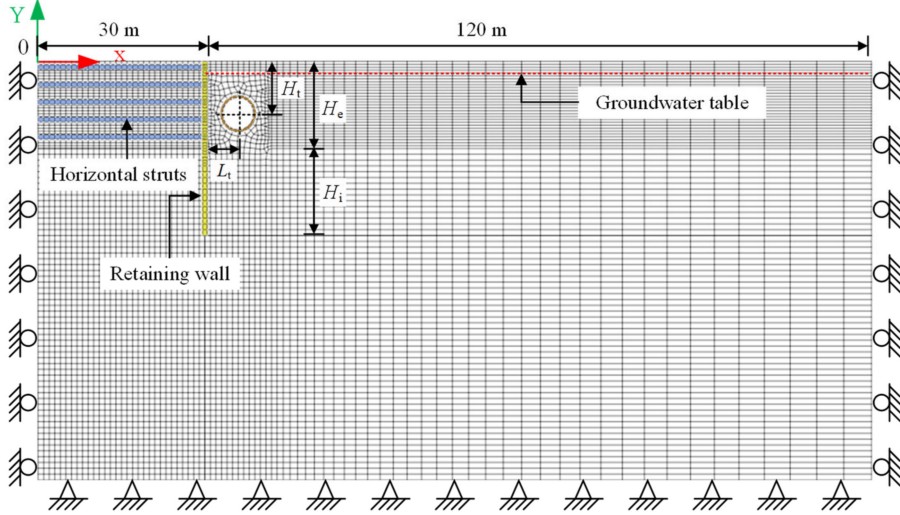

**Figure 1.** FEM numerical model used to study the effect of adjacent excavation on existing tunnel.

In the model, the soil is simulated using a plane strain element, the retaining wall, horizontal struts and existing tunnel are simulated using beam elements, and the contacts of the wall–soil and tunnel–soil are simulated using the interface element. For displacement boundary conditions, the left and right boundaries are limited to the horizontal displacement, the bottom boundary is limited to the horizontal and vertical displacements, and the top boundary is kept free. For load boundary conditions, only the self-weight stress is considered, and no other loads are considered. For water level boundary conditions, the initial groundwater table is 1.5 m below the ground surface, and the pressure head on the groundwater line is 0.

### 2.1.2. Constitutive Model and Parameters

Some studies [15–22] have demonstrated that the hardening soil model with small strain stiff (HSS) is an advanced constitutive model for soil in numerical simulations related to excavation issues, and the simulation results using it can obtain reasonable retaining structure and soil displacements. Therefore, the HSS model for soil was used in this study. In addition to the basic physical parameters such as the unit weight $\gamma$, void ratio $e$, the HSS model includes 13 constitutive model parameters: $c'$ is the effective cohesive force, $\varphi'$ is the effective internal friction angle, $\psi$ is the dilatancy angle, $E_{50}^{ref}$ is the secant stiffness in standard drained triaxial test, $E_{oed}^{ref}$ is the tangent stiffness for oedometer primary loading, $E_{ur}^{ref}$ is the triaxial unloading-reloading stiffness, $G_0^{ref}$ is the small stain shear stiffness modulus, $\gamma_{0.7}$ is the shear strain amplitude at 0.7 $G_0^{ref}$, $m$ is the power for the stress-level dependency of stiffness, $\nu_{ur}$ is the Poisson's ratio for unloading-reloading, $P^{ref}$ is the reference stress for stiffness, $R_f$ is the failure ratio, and $K_0$ is the stress ratio of the horizontal effective stress to the vertical effective stress in a normally consolidated state.

Liu et al. (2022) [23] collected 42 case histories in which the existing tunnel was affected by adjacent deep excavation and found that the cases that occurred in the soft soil represented by silty clay occupy a dominant position in actual engineering. Therefore, silty clay was chosen as the soil for calculation in the model, and it is assumed to be a single homogeneous layer to eliminate the influence of soil stratification. As this study is an extension of the study by Zheng et al. (2016, 2017) [11,12], the parameters of the HSS model for the soil are the same, as listed in Table 1.

**Table 1.** Physical and mechanical parameters of HSS model for the soil [11,12].

| No. | Parameters | Values | No. | Parameters | Values |
|---|---|---|---|---|---|
| 1 | $\gamma$ | 19.8 kN/m$^3$ | 9 | $m$ | 0.8 |
| 2 | $e$ | 0.6 | 10 | $\nu_{ur}$ | 0.2 |
| 3 | $c'$ | 14.0 kPa | 11 | $p^{ref}$ | 100 kPa |
| 4 | $\varphi'$ | 25.7° | 12 | $R_f$ | 0.9 |
| 5 | $\psi$ | 0 | 13 | $K_0$ | 0.57 |
| 6 | $E_{50}^{ref}$ | 7.2 MPa | 14 | $G_0^{ref}$ | 99.3 MPa |
| 7 | $E_{oed}^{ref}$ | 5.1 MPa | 15 | $\gamma_{0.7}$ | $0.20 \times 10^{-3}$ |
| 8 | $E_{ur}^{ref}$ | 36.8 MPa | | | |

The retaining wall, horizontal struts, and existing tunnel adopt the linear elastic constitutive model. The unit weight is $\gamma$ = 24.5 kN/m$^3$, the elastic modulus is $E$ = 30 GPa, and the Poisson's ratio is $\nu$ = 0.2. In the model, the shield tunnel is regarded as a homogeneous ring without transverse and longitudinal joints but, actually, the tunnel segments are assembled by blots. Therefore, the effective stiffness ratio of tunnel $\eta$ ($\eta \leq 1$) is introduced to reflect the weakening effect of segment joints on the tunnel stiffness, i.e., the equivalent stiffness of the tunnel is $\eta EI$, where $EI$ is the bending stiffness of the homogeneous tunnel before stiffness reduction. Some studies [24–28] demonstrated that the value of $\eta$ is related to the soil conditions, the form of the segment joints, and the manner of segment assembly. In this study, the value of $\eta$ = 0.75 is adopted, referring to the above studies.

The contacts of the wall–soil and tunnel–soil adopt the interface model. The main nonlinear parameters of the interface model are the normal stiffness modulus ($K_n$) and shear stiffness modulus ($K_t$), which can be determined as:

$$K_n = E_{oed,i}/t_v \tag{1}$$

$$K_t = G_i/t_v \tag{2}$$

where $E_{oed,i} = 2G_i(1 - \nu_i)/(1 - 2\nu_i)$; $G_i = R \times G_{soil}$, and $G_{soil} = E/2(1 + \nu_{soil})$; $\nu_i$ is the Poisson's ratio of interface, and $\nu_i = 0.45$ is adopted; $\nu_{soil}$ is the Poisson's ratio of soil, and $\nu_{soil} = 0.35$ is adopted; $t_v$ is the virtual thickness coefficient of interface, which is generally in the range of 0.01~0.1, and $t_v = 0.1$ is adopted for concrete wall–soil interface; $R$ is the strength reduction coefficient, which is generally in the range of 0.7~1.0, and $R = 0.7$ is adopted for concrete wall-soil interface.

### 2.1.3. Simulation Conditions

In the numerical simulation, the factors considered include the excavation depth ($H_e$), the maximum deflection of the retaining wall ($\delta_{hm}$), the deformation mode of the retaining wall, and the positions of the tunnel ($H_t$, $L_t$).

For the excavation depth, $H_e$ = 18 m is adopted. For the deformation magnitude of the retaining wall, the guideline CCES 03–2016 [29] provides the 3-level control standards of the retaining wall deflection in soft soil, i.e., 0.18%$H_e$, 0.3%$H_e$ and 0.7%$H_e$. Considering that the deformation of braced deep excavation is usually controlled to a good level, so $\delta_{hm} = 0.18\%H_e$ is adopted. For the deformation mode of the retaining wall, the convex type is adopted. The tunnel position is set using the following rules: in the horizontal direction, the spacing between each tunnel center is 3 m when $L_t \leq H_e$, and that is 6 m

when $L_t > H_e$, and the maximum value of $L_t = 3.0H_e$; in the vertical direction, the spacing of each tunnel center is 3 m when $H_t \leq H_e$, and that is 6 m when $H_t > H_e$, and the maximum value of $H_t = 3.0H_e$. According to the statistics of 42 case histories by Liu et al. (2022) [23], the adopted $L_t$ and $H_t$ cover the majority of tunnel positions in actual engineering and are widely representative. Because no construction activities are allowed within 3 m of the edge of the tunnel, and the buried depth of the tunnel crown should not be less than 1.0 time the diameter of the tunnel, the value of $L_t$ starts from 6 m, and the value of $H_t$ starts from 9 m. After considering the above factors, a total of 110 numerical models are established, as listed in Table 2.

**Table 2.** Numerical simulation conditions.

| $H_e$ (m) | $\delta_{hm}$ (m) | Deformation Mode of DW | $H_t$ (m) | $L_t$ (m) | Number |
|---|---|---|---|---|---|
| 18 | 0.18%$H_e$ | convex type | 9, 12, 15, 18, 24, 30, 36, 42, 48, 54 | 6, 9, 12, 15, 18, 24, 30, 36, 42, 48, 54 | 110 |

### 2.2. Deformation Behaviors of Adjacent Tunnel

Figure 2a shows the maximum horizontal displacement of tunnels at different positions outside the pit, in which a negative value indicates the displacement towards the pit. It is observed, when $L_t/H_e \leq 1.0$, that the deep excavation has a large effect on the tunnel, and the horizontal displacement curves appear to be a convex type. Instead, when $L_t/H_e > 1.0$, the effect of deep excavation on the tunnel obviously decreases, and the horizontal displacement curves appear to be a linear type.

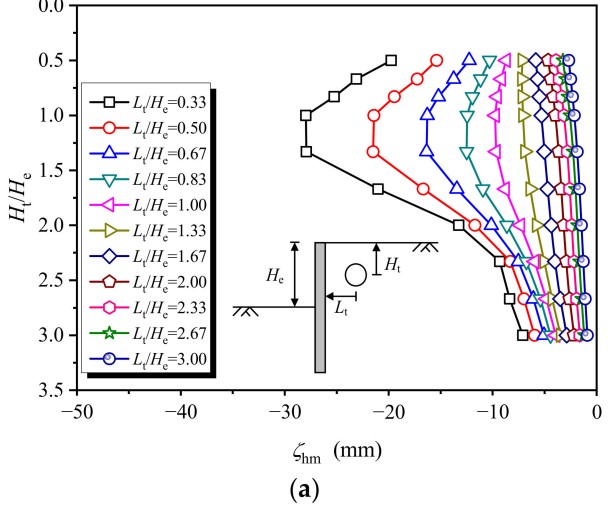

(**a**)

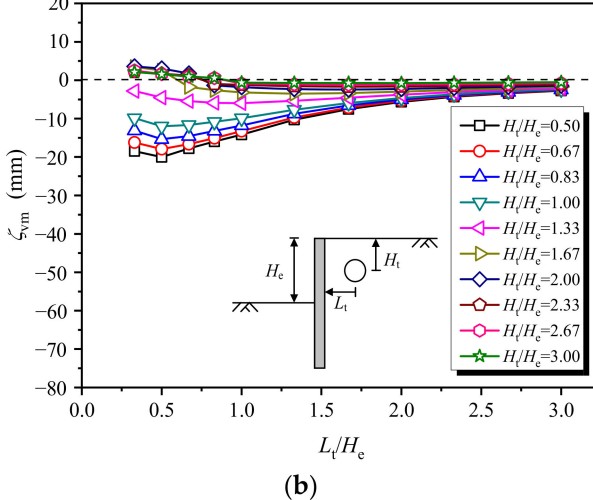

(**b**)

**Figure 2.** The maximum displacement of tunnels at different positions: (**a**) horizontal displacement; (**b**) vertical displacement.

Figure 2b shows the maximum vertical displacement of tunnels at different positions outside the pit, in which a negative value indicates the settlement, and a positive value indicates the heave. It is observed, when $H_t/H_e \leq 1.33$, that tunnel settlement occurs, and the shallower the buried depth, the larger the settlement that occurs. Additionally, when $H_t/H_e > 1.33$ and $L_t/H_e \leq 0.83$, tunnel heave occurs, which is induced by the excavation unloading.

### 2.3. Division of Influenced Zone of Deep Excavation on Adjacent Tunnel

Figure 3 shows the isoline of tunnel horizontal and vertical displacements, in which the tunnel position is normalized by the excavation depth, and the horizontal and vertical ordinates are $L_t/H_e$ and $H_t/H_e$, respectively. Excluding the scope not allowing construction,

the value of $L_t/H_e$ starts from 0.33, and the value of $H_t/H_e$ starts from 0.5. The isoline presents the horizontal and vertical displacements of the tunnel at different positions outside the pit and, on this basis, the influenced zone of deep excavation on adjacent tunnels can be divided corresponding to different tunnel displacement control standards.

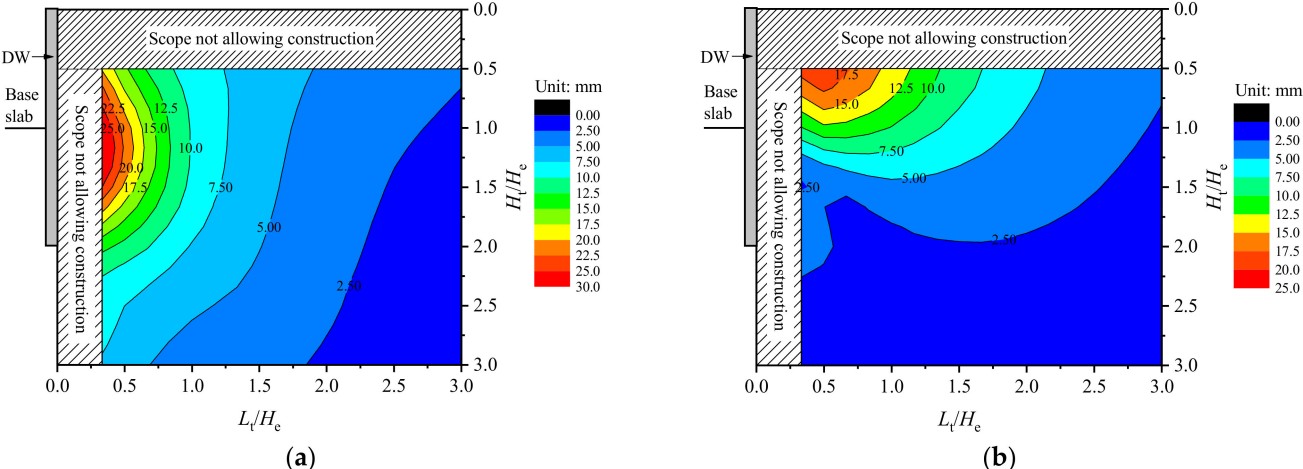

**Figure 3.** Isolines of tunnel displacements: (**a**) horizontal displacement isoline; (**b**) vertical displacement isoline.

The code CJJ/T 202–2013 [30] provides an early warning value of 10 mm and a control value of 20 mm for the horizontal and vertical displacements of the tunnel, respectively. Additionally, the code GB 50911–2013 [9] requires that the settlement of the tunnel should be controlled within 3–10 mm, the tunnel heave should be controlled within 3–5 mm, and the horizontal displacement of the tunnel should be controlled within 3–5 mm. Here, based on the requirements of the abovementioned codes, the maximum tunnel displacement of 20 mm, 10 mm, and 5 mm were selected as the 3-level tunnel displacement control standards. Then, according to the 3-level standards, determining their corresponding isoline range in Figure 3, i.e., the influenced zone, as shown in Figure 4.

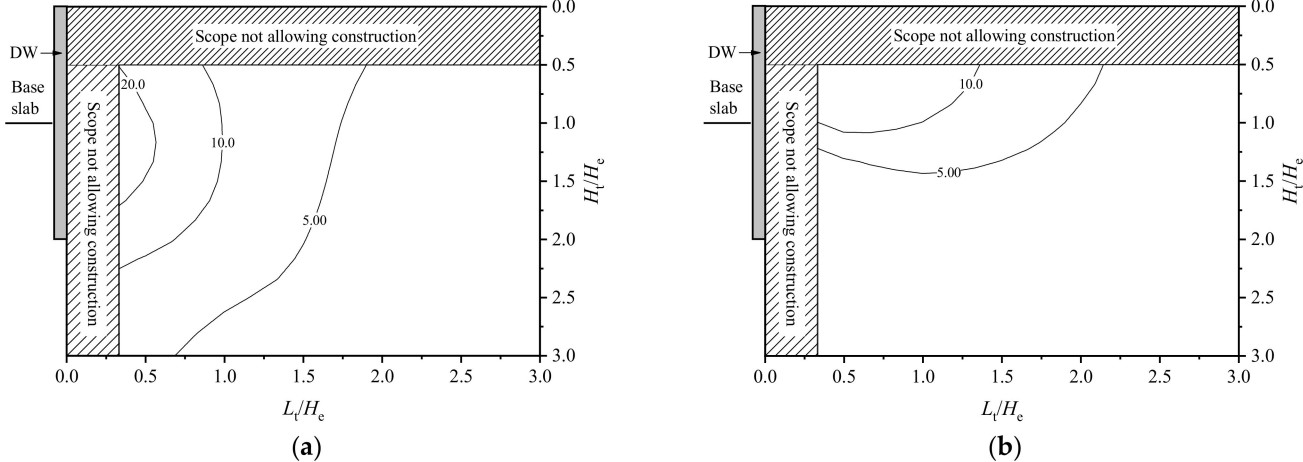

**Figure 4.** Influenced zones for tunnel displacements: (**a**) influenced zone for tunnel horizontal displacement; (**b**) influenced zone for tunnel vertical displacement.

Next, overlay the influenced zones for the horizontal and vertical displacements together, and draw the envelope line of the influenced zones corresponding to the tunnel displacement control standards of 20 mm, 10 mm, and 5 mm, and thus the preliminary division result of the influenced zones is obtained, as shown in Figure 5. Both horizontal and vertical displacements of the tunnel are considered in the envelope, i.e., at least one

of the horizontal or vertical displacements of the tunnel within the envelope exceeds the tunnel displacement control standards.

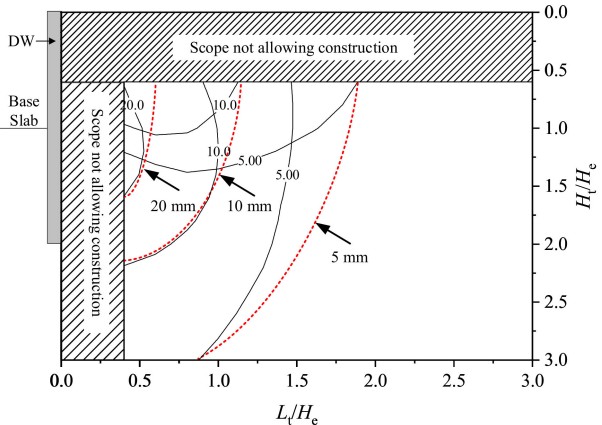

**Figure 5.** Influenced zones of deep excavation on adjacent tunnel displacement.

To characterize the scope of the influenced zone, according to its features, the influenced zone in Figure 5 is simplified as a right-angled trapezoid, and the isoline of tunnel displacement is simplified as a polyline. The three polylines represent the boundaries of tunnel displacements of 20 mm, 10 mm, and 5 mm, respectively. Referencing the study conducted by Zheng et al. (2016, 2017) [11,12], the range outside the pit is divided into: I—primary influenced zone, II—secondary influenced zone, III—general influenced zone, and IV—weak influenced zone, as shown in Figure 6.

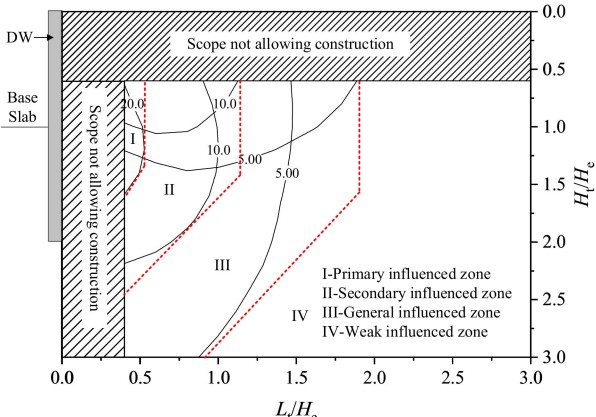

**Figure 6.** Influenced zones of deep excavation on adjacent tunnel displacement after simplification.

To quantitatively determine the scope of the influenced zones shown in Figure 6, the coordinates of three points on the polylines are defined: the width coefficient $M$, the depth coefficient $N_1$, and the depth coefficient $N_2$, as shown in Figure 7. In this instance, $M$ reflects the influence potential of deep excavation on the tunnel displacement in the horizontal direction, $N_2$ reflects the influence potential of deep excavation on the tunnel displacement in the vertical direction, and the larger the above values, the greater the influence degree indicated. Using these three coordinates, the scope of the influenced zones can be quickly determined.

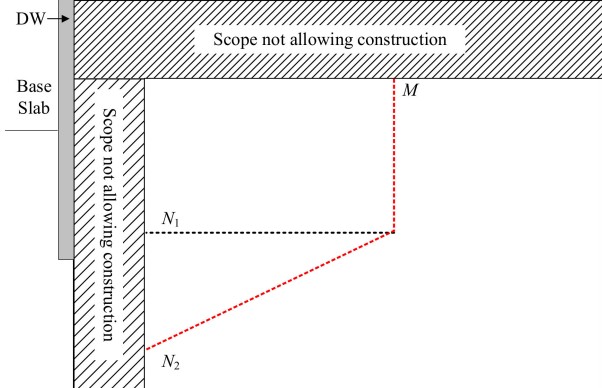

**Figure 7.** Mode and quantitative determination method of influenced zone.

The values of $M$, $N_1$, and $N_2$ corresponding to the 3-level tunnel displacement control standards of 20 mm, 10 mm, and 5 mm are listed in Table 3.

**Table 3.** Determination parameters of influenced zones.

| 20 mm | | | 10 mm | | | 5 mm | | |
|---|---|---|---|---|---|---|---|---|
| $M$ | $N_1$ | $N_2$ | $M$ | $N_1$ | $N_2$ | $M$ | $N_1$ | $N_2$ |
| 0.60 | 1.30 | 1.70 | 1.35 | 1.30 | 2.35 | 2.15 | 1.5 | 3.30 |

## 3. Control Effect of Ground Improvement on Tunnel Displacement

### 3.1. Control Measures of Tunnel Displacement

By the collection and statistics of 42 case histories in which the tunnel was affected by adjacent deep excavation, Liu et al. (2022) [23] found the commonly used control measures of tunnel displacement mainly include ground improvement, zoned excavation, isolation pile, and field monitoring in actual engineering. Figure 8 shows the number and proportion of each control measure under different soil conditions. It is observed that the soil condition has a significant effect on the choice of tunnel displacement control measures. The control measures used in sandy pebble and weathered rock strata with good properties are much less than those used in soft clay, silt, and silty sand strata with poor properties. In soft clay strata, except for field monitoring, ground improvement is the most widely used control measure. Therefore, ground improvement was chosen to study the control effects on the tunnel displacement and influenced zone of deep excavation.

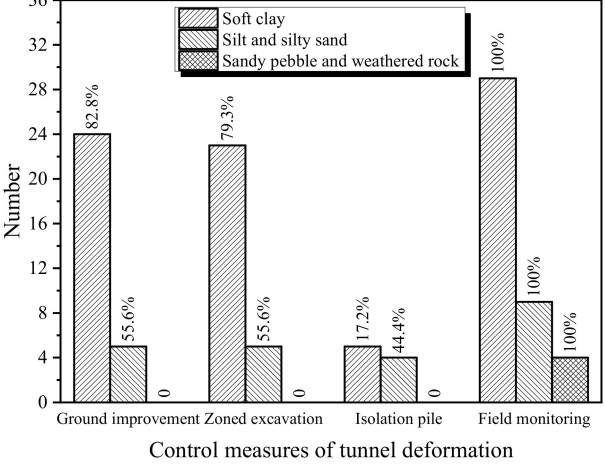

**Figure 8.** Control measures of tunnel displacement under different soil conditions.

### 3.2. Evaluation Index of Control Effect on Tunnel Displacement

Figure 9 shows the schematic diagram of ground improvement outside the pit, in which the excavation depth is $H_e$, the buried depth of the tunnel center is $H_t$, the horizontal distance from the tunnel center to the retaining wall is $L_t$, and the ground improvement is between the retaining wall and existing tunnel. According to the statistics of case histories by Liu et al. (2022) [23], in actual engineering the tunnel position $H_t/H_e = 0.8 \sim 1.0$ and $L_t/H_e = 0.5 \sim 1.0$ accounts for a high proportion. Therefore, in this study, $H_e = 18$ m, $H_t = H_e$, and $L_t = H_e$ were adopted to analyze the control effects of ground improvement on the tunnel displacement and influenced zone of deep excavation.

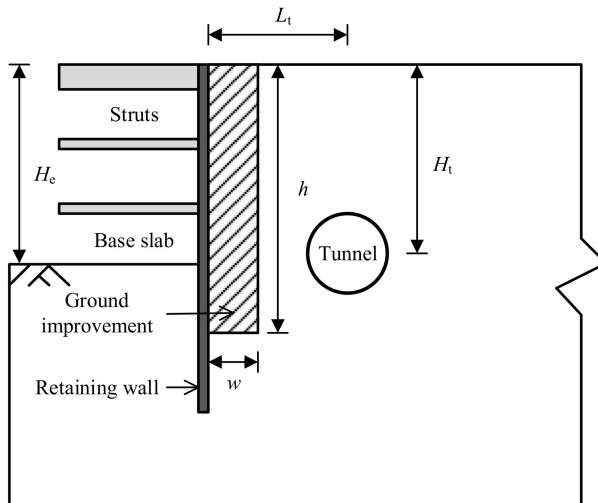

**Figure 9.** Schematic diagram of ground improvement outside the pit.

For the tunnel affected by adjacent deep excavation, it is important to control the horizontal displacement [23], so this study mainly discusses the control effect of ground improvement on the horizontal displacement of the tunnel. An index of displacement control effectiveness ($\eta_h$) is defined to quantitively characterize the control effect on tunnel displacement as:

$$\eta_h = \frac{\zeta_{hm} - \zeta'_{hm}}{\zeta_{hm}} \tag{3}$$

where $\zeta_{hm}$ is the maximum horizontal displacement of the tunnel before ground improvement, and $\zeta'_{hm}$ is the maximum horizontal displacement of the tunnel after ground improvement. If $\eta_h < 0$, it means that the ground improvement increases the displacement of the tunnel, and has a negative effect on the control of the tunnel displacement; if $\eta_h = 0$, it means that the ground improvement is ineffective in controlling the tunnel displacement; if $0 < \eta_h < 1$, it means that the ground improvement is effective in controlling the tunnel displacement, and the greater $\eta_h$ is, the better the control effect is.

### 3.3. Simulation Conditions of Ground Improvement

In this study, numerical simulations were conducted to investigate the effects of ground improvement strength ($q_u$), ground improvement depth ($h$), and ground improvement width ($w$) on the control effect on tunnel displacement.

Gong (2008) [31] pointed out, for cement-improved soil, the unconfined compressive strength $q_u = 0.5 \sim 4$ Mpa, the ratio of cohesive force to compressive strength $c/q_u = 0.2 \sim 0.3$, the internal friction angle $\varphi = 20 \sim 30°$, and the deformation modulus $E = (120 \sim 150) \, q_u$. Liu and Wang (2009) [32] pointed out, for cement-improved soil, when the unconfined compressive strength $q_u = 0.5 \sim 4.0$ MPa, its cohesive force $c = 0.1 \sim 1.1$ MPa, the internal friction angle $\varphi = 20 \sim 30°$, and the deformation modulus $E = (100 \sim 150) \, q_u$.

Referring to the above research, in numerical simulations, the unconfined compressive strengths of improved soil $q_u$ = 0.5 MPa, 1.0 MPa, 1.5 MPa, 2.0 MPa, 2.5 MPa, 3.0 MPa, 3.5 MPa, and 4.0 MPa are adopted, totaling eight values. The ratio of cohesive force to unconfined compressive strength $c/q_u$ = 0.25 is adopted; the internal friction angle $\varphi$ is determined at equal intervals within 20~30° according to different unconfined compressive strengths; the deformation modulus $E = 130\,q_u$ is adopted. The basic physical parameters of cement-improved soil, such as gravity $\gamma$ and Poisson's ratio $\nu$, are taken at equal intervals between 20~21.4 kN/m$^3$ and 0.20~0.34 according to different unconfined compressive strengths. Table 4 lists the physical and mechanical parameters of cement-improved soil, corresponding to different unconfined compressive strengths.

**Table 4.** Physical and mechanical parameters of cement-improved soil.

| $q_u$ (MPa) | $c$ (kPa) | $\varphi$ (°) | $E$ (MPa) | $\gamma$ (kN/m$^3$) | $\nu$ |
|---|---|---|---|---|---|
| 0.5 | 125 | 21 | 65 | 20.0 | 0.34 |
| 1 | 250 | 22 | 130 | 20.2 | 0.32 |
| 1.5 | 375 | 23 | 195 | 20.4 | 0.30 |
| 2 | 500 | 24 | 260 | 20.6 | 0.28 |
| 2.5 | 625 | 25 | 325 | 20.8 | 0.26 |
| 3 | 750 | 26 | 390 | 21.0 | 0.24 |
| 3.5 | 875 | 27 | 455 | 21.2 | 0.22 |
| 4 | 1000 | 28 | 520 | 21.4 | 0.20 |

To block the propagation path of lateral unloading stress induced by deep excavation, the ground improvement depth outside the pit is generally deeper than the buried depth of the tunnel. In this study, the values of $h = 1.5H_t$, $2.0H_t$, $2.5H_t$, and $3.0H_t$ are adopted, totaling four values.

The ground improvement width is determined according to the horizontal distance between the tunnel and the retaining wall, and the distance from the improved soil to the springing line of the tunnel should be less than 1 m to avoid construction disturbance to the tunnel. In this study, the values of $w = L_t/6$, $L_t/3$, $L_t/2$, and $2L_t/3$ are adopted, totaling four values.

After considering the three factors of ground improvement strength $q_u$, ground improvement depth $h$, and ground improvement width $w$, a total of 128 simulation conditions and numerical models are set.

### 3.4. Effect of Ground Improvement on Tunnel Displacement
3.4.1. Effect of Ground Improvement Strength

Figure 10 shows the evolution curves of $\eta_h$ with $q_u$. It is observed that under the four conditions with different $h$ values, $\eta_h$ increases nonlinearly with $q_u$, and the larger $w$ is, the greater the increase in $\eta_h$ that is obtained. Under the condition of $h = 1.5H_t$, $\eta_h$ increases slightly with $q_u$, and it has a maximum increase of 13% when $q_u$ increases from 0.5 MPa to 4.0 MPa. Under the conditions of $h = 2.0H_t$, $h = 2.5H_t$, and $h = 3.0H_t$, $\eta_h$ increases obviously before $q_u$ = 1.5 MPa and then gradually slows down, and it has a maximum increase of 40%, 44%, and 45% when $q_u$ increases from 0.5 MPa to 4.0 MPa.

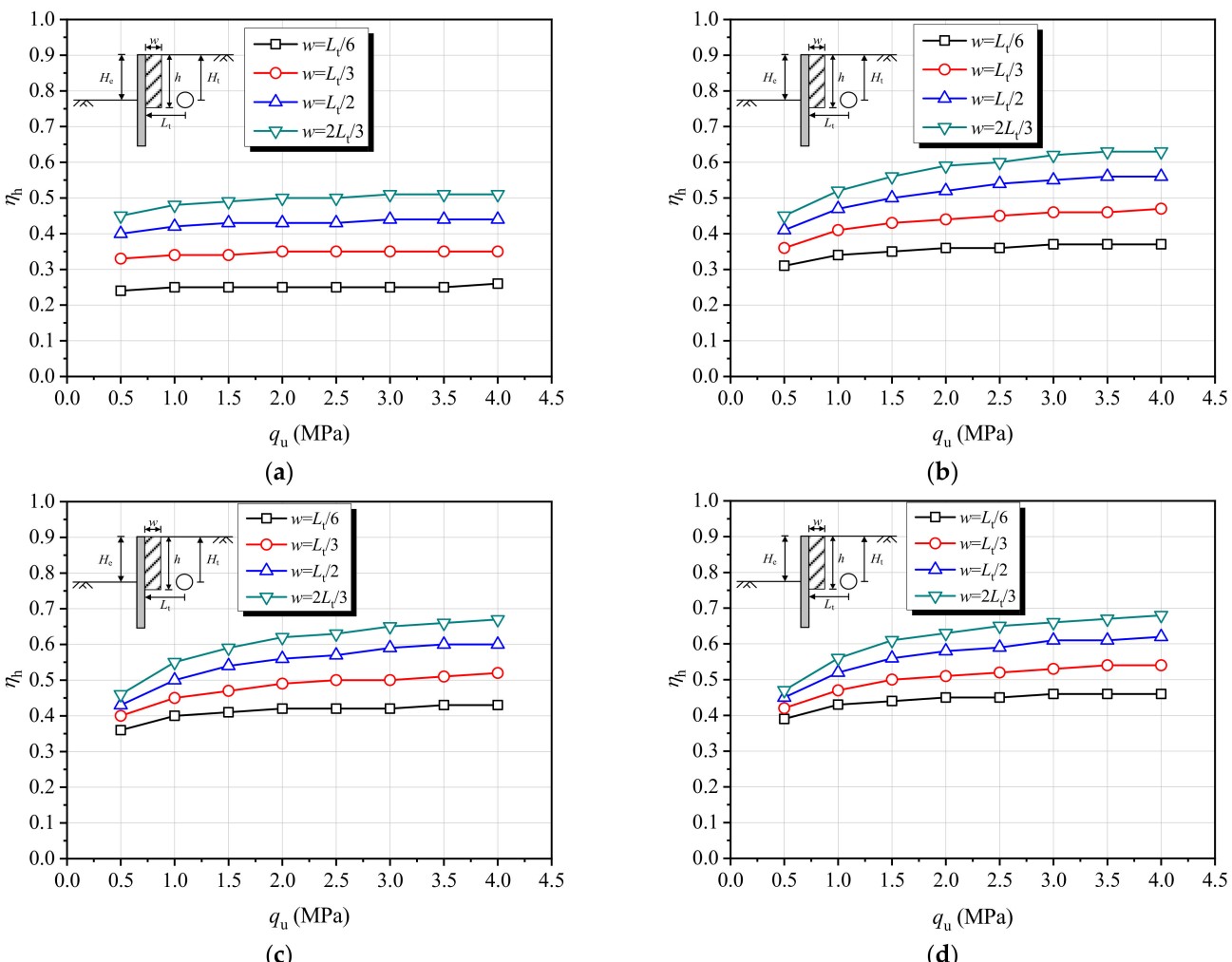

**Figure 10.** Evolution curves of $\eta_h$ with $q_u$: (**a**) $h = 1.5H_t$; (**b**) $h = 2.0H_t$; (**c**) $h = 2.5H_t$; (**d**) $h = 3.0H_t$.

It is concluded from the above analysis that there are limits to increasing the control effect on tunnel displacement by only increasing the ground improvement strength. Considering the control effect and engineering economy together, it is suggested that the ground improvement strength should be kept within 1.5~2 MPa.

### 3.4.2. Effect of Ground Improvement Depth

Figure 11 shows the evolution curves of $\eta_h$ with $h$. It is observed that under the four conditions with different $w$ values, $\eta_h$ increases nonlinearly with $h$, and the larger $q_u$ is, the greater the increase in $\eta_h$ that is obtained. The characteristics of the curve show that $\eta_h$ increases obviously before $h = 2.0H_t$ and slows down obviously after $h = 2.0H_t$, which may be related to the relative position of the improved soil in the displacement field outside the pit.

As shown in Figure 12, the excavation-induced unloading makes the soil outside the pit move towards the pit, forming a potential "sliding surface". The soil displacement above the sliding surface is greater than that below the sliding surface, and the toe of the sliding surface is near the pit bottom elevation. If the ground improvement depth $h > 1.0H_t$, the improved soil below the sliding surface plays an embedded role, which is similar to an anti-slide pile and can block the displacement of the soil outside the pit. The deeper the improved soil embedded below the sliding surface is, the better the barrier effect it plays, and a better control effect on tunnel displacement is obtained. However, when the embedded depth is greater than $1.0H_t$, that is, the ground improvement depth $h > 2H_t$, the

contribution of the embedded depth to the barrier effect decreases, and the increasing trend of $\eta_h$ slows down with $h$.

**Figure 11.** Evolution curves of $\eta_h$ with $h$: (**a**) $w = L_t/6$; (**b**) $w = L_t/3$; (**c**) $w = L_t/2$; (**d**) $w = 2L_t/3$.

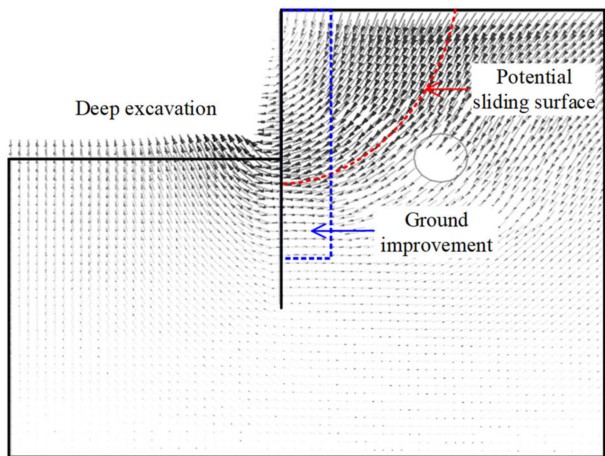

**Figure 12.** Relative positions between ground improvement and potential sliding surface.

It is concluded from the above analysis, that an optimal value exists for the ground improvement depth outside the pit. If it exceeds this value, the contribution of the ground

improvement depth to the tunnel displacement control effect will be weakened. Considering the control effect and engineering economy, it is suggested that the ground improvement depth $h = 2H_e$. For this example, because $H_t = H_e$, it is suggested that the ground improvement depth $h = 2H_t$.

### 3.4.3. Effect of Ground Improvement Width

Figure 13 shows the evolution curves of $\eta_h$ with $w$. It is observed that under the four conditions with different $h$ values, $\eta_h$ increases linearly with $w$, and the greater the ground improvement strength $q_u$ is, the more obvious the increase of $\eta_h$ is obtained. Under the condition of $h = 1.5H_t$, when the $w$ increases from $L_t/6$ to $2L_t/3$, the $\eta_h$ value has a maximum increase of 100%. Under the conditions of $h = 2.0H_t$, $h = 2.5H_t$ and $h = 3.0H_t$, when $w$ increases from $L_t/6$ to $2L_t/3$, the $\eta_h$ value has a maximum increase of 70%, 55%, and 48%, respectively.

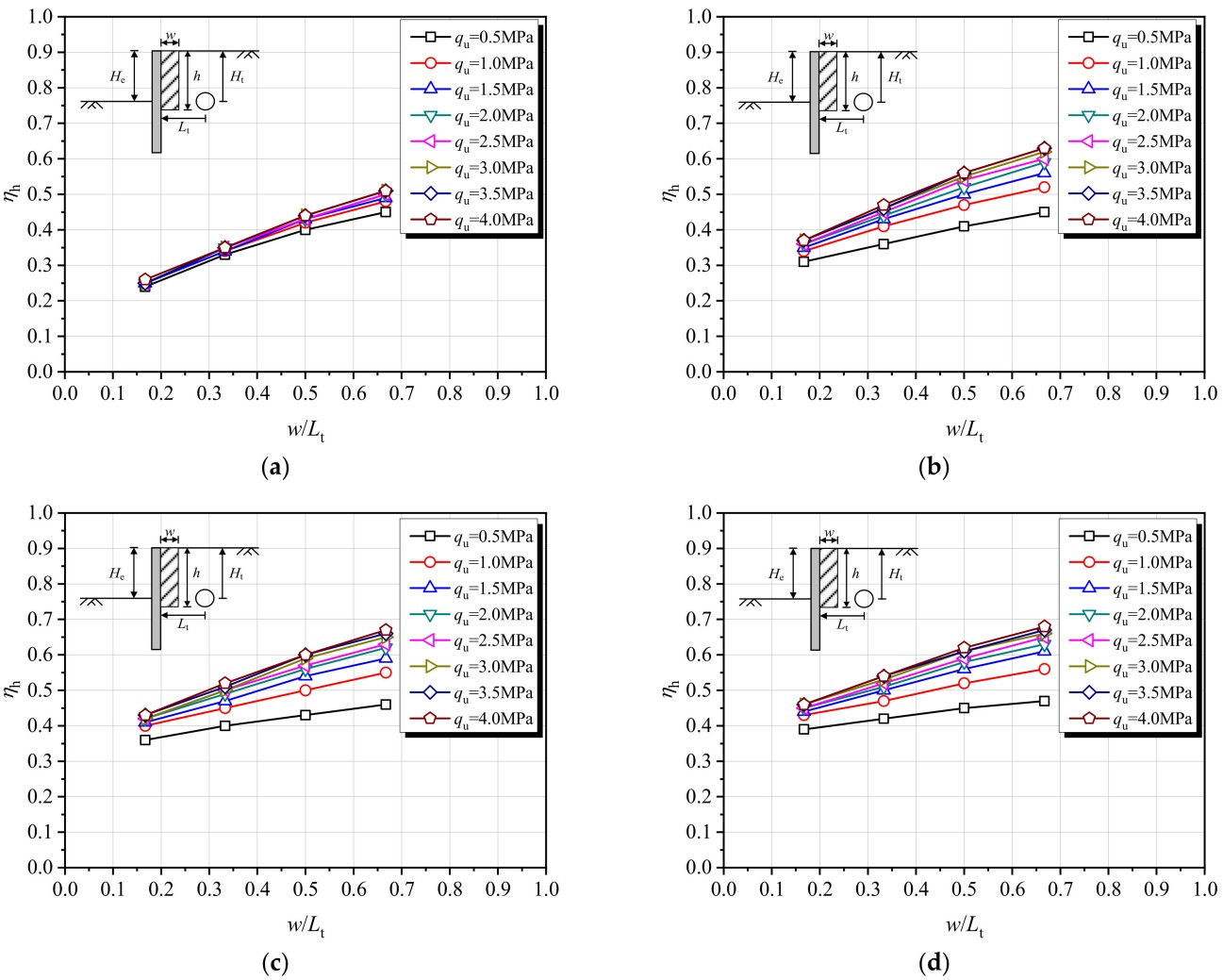

**Figure 13.** Evolution curves of $\eta_h$ with $w$: (**a**) $h = 1.5H_t$; (**b**) $h = 2.0H_t$; (**c**) $h = 2.5H_t$; (**d**) $h = 3.0H_t$.

It is concluded from the above analysis that the control effect on tunnel displacement can be increased by increasing the ground improvement width. In actual engineering, if there is enough space between the deep excavation and the tunnel to allow ground improvement and ensure the safety of the tunnel, it is suggested to increase the ground improvement width as much as possible.

## 4. Control Effect of Ground Improvement on Influenced Zone of Deep Excavation

### 4.1. Effect of Ground Improvement on Tunnel Displacement Distribution Behaviors

It is known from the above analysis that the ground improvement outside the pit has a positive effect on the tunnel displacement induced by deep excavation. Considering the control effect and engineering economy, it is suggested that the ground improvement strength should be kept within 1.5~2 MPa, the ground improvement depth should be $2H_e$, and the ground improvement width should be increased as much as possible if the site condition allows. In this study, the ground improvement strength of 2 MPa, the ground improvement depth of $2H_e$, and the ground improvement width extending to 1 m to the edge of the tunnel were adopted to analyze the effect of ground improvement on the displacement distribution characteristics of the tunnel and the scope of the influenced zone.

Figure 14 shows the horizontal and vertical displacement curves of tunnels at different positions outside the pit after ground improvement. To facilitate comparison with the results under the conditions without ground improvement, the coordinates are the same as those in Figure 2. It is observed that, after ground improvement, the displacement has an obvious decrease. Before ground improvement, the maximum horizontal displacement and settlement of the tunnel are 28 mm and 20 mm. After ground improvement, the maximum horizontal displacement and settlement of the tunnel are 14 mm and 6.5 mm, which is reduced by 50% and 67.5%, respectively. It means that ground improvement outside the pit can effectively control both the horizontal and vertical displacements of the tunnel.

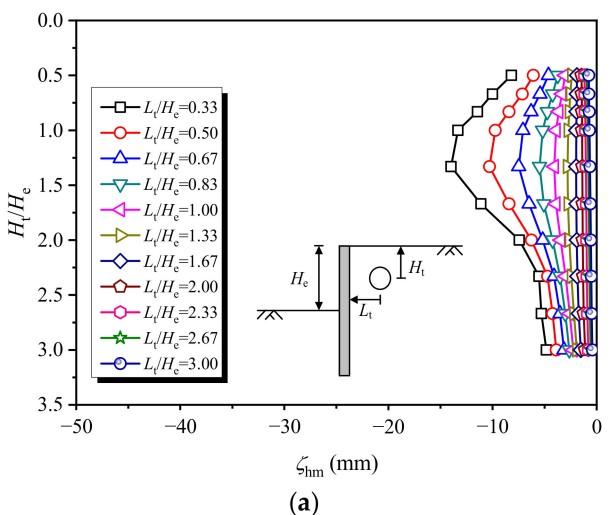

(**a**)

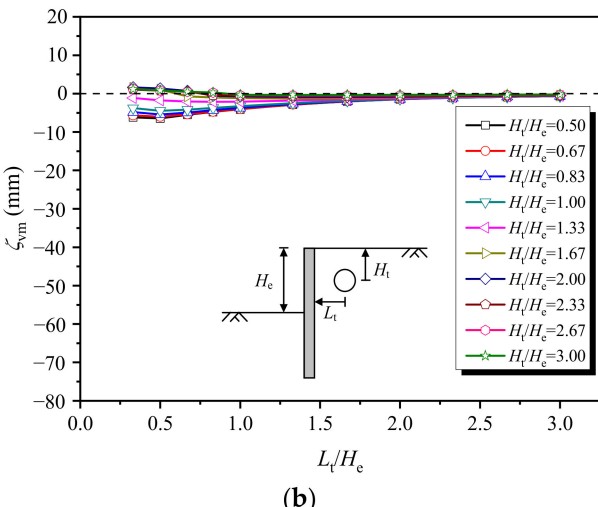

(**b**)

**Figure 14.** The maximum displacement of tunnels at different positions after ground improvement: (**a**) horizontal displacement; (**b**) vertical displacement.

### 4.2. Effect of Ground Improvement on Influenced Zone of Deep Excavation

The ground improvement outside the pit changes the displacement characteristics of the existing tunnel, so it inevitably changes the influenced zone of deep excavation on tunnel displacement. The following will focus on the redistribution of the influenced zone after ground improvement. The method for dividing the influenced zone is consistent with that of Section 2.3. By setting the different positions of the tunnel outside the pit, numerical simulations are adopted to calculate the tunnel displacement induced by excavation. On this basis, through the analysis of tunnel displacement isoline, combined with 3-level tunnel displacement control standards, the scope of the influenced zone corresponding to each control standard is divided.

Figure 15 shows the horizontal and vertical displacement isolines of the tunnel at different positions after ground improvement. Compared with the results in Figure 3 without ground improvement, the tunnel displacement decreases obviously with the maximum horizontal displacement not exceeding 15 mm and the maximum vertical displacement

not exceeding 10 mm, and the scope of the isoline corresponding to the same displacement control standard decreases.

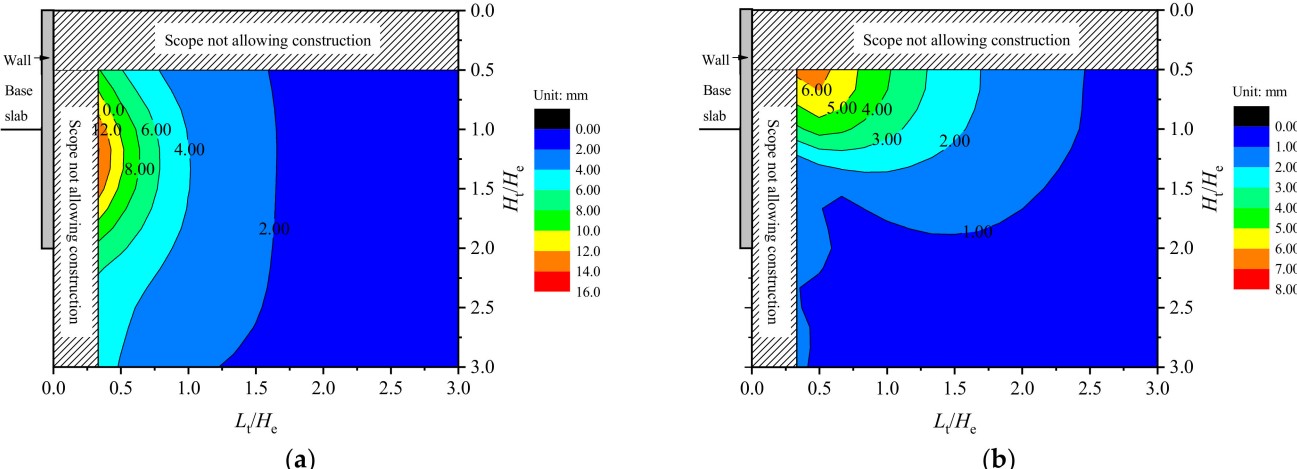

**Figure 15.** Isolines of tunnel displacement after ground improvement: (**a**) horizontal displacement isoline; (**b**) vertical displacement isoline.

Then, according to the division method for the influence zone of deep excavation on adjacent tunnel displacement in Section 2.3, the influenced zone corresponding to the 3-level tunnel displacement control standards is given after ground improvement. The range outside the pit is divided into: II—secondary influenced zone, III—general influenced zone, and IV—weak influenced zone, as shown in Figure 16. The determination parameters of the scope of the influenced zone after ground improvement are listed in Table 5.

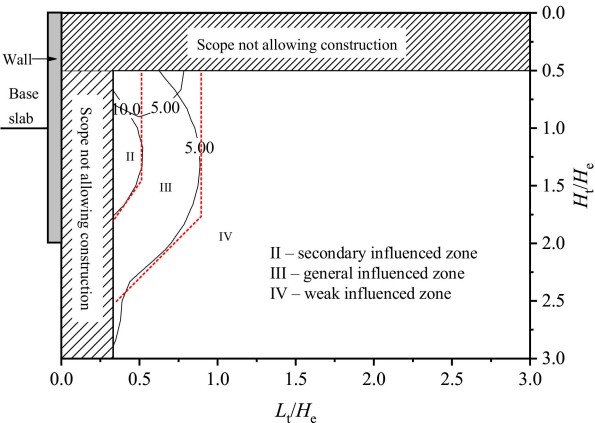

**Figure 16.** Influenced zones of deep excavation on adjacent tunnel displacement after ground improvement.

**Table 5.** Determination parameters of the scope of influenced zones after ground improvement.

| | 20 mm | | | 10 mm | | | 5 mm | |
|---|---|---|---|---|---|---|---|---|
| $M$ | $N_1$ | $N_2$ | $M$ | $N_1$ | $N_2$ | $M$ | $N_1$ | $N_2$ |
| / | / | / | 0.55 | 1.45 | 1.75 | 0.90 | 1.75 | 2.50 |

Compared with the division results of the influenced zone without ground improvement in Table 3, the scope of the influenced zone is significantly reduced, and the I—primary influenced zone corresponding to the tunnel displacement control standard of 20 mm no longer exists; the width and depth of the II—secondary influenced zone is reduced by 60% and 30%, respectively. The width and depth of III—general influenced zone are reduced

by 60% and 26%, respectively. Therefore, it is concluded that the ground improvement outside the pit plays an important role in reducing the influenced zone of deep excavation on adjacent tunnel displacement.

## 5. Discussion

The influenced zone of deep excavation on adjacent tunnel displacement and the control effect of ground improvement are influenced by many factors, such as geological conditions, structure forms, and construction workmanship. In this study, the braced deep excavation with a depth of 18 m and shield tunnel with a diameter of 6 m in thick soft soil was chosen for study, and some preliminary results were obtained. For deep excavations and tunnels under other conditions, the results need to be further discussed and verified.

## 6. Conclusions

Based on the current research, numerical simulations were conducted to analyze the tunnel displacement behaviors induced by adjacent deep excavation, considering different tunnel positions outside the pit. On this basis, the influenced zone of deep excavation on adjacent tunnel displacement was divided. Then, the commonly used control measure of ground improvement was chosen to study the effect of strength, depth, and width of improved soil outside the pit on the tunnel displacement. An index of displacement control effectiveness ($\eta$) was proposed to quantitively characterize the control effect on tunnel displacement. Considering the control effect and engineering economy, the suggested values of strength, depth, and width of the ground improvement were provided. Finally, the control effect of ground improvement outside the pit on the influenced zone of deep excavation was evaluated using the suggested construction parameters. The conclusions obtained are as follows:

(1) Based on the displacement behaviors of the tunnel induced by deep excavation, combined with the 3-level tunnel displacement control standards of 20 mm, 10 mm, and 5 mm, the range outside the pit can be divided into: I—primary influenced zone, II—secondary influenced zone, III—general influenced zone, and IV—weak influenced zone.

(2) There are limits to increasing the control effect on tunnel displacement by only increasing the ground improvement strength. Considering the control effect and engineering economy together, it is suggested that the ground improvement strength should be kept within 1.5~2 MPa.

(3) An optimal value exists for the ground improvement depth outside the pit. If it exceeds this value, the contribution of the ground improvement depth to the tunnel displacement control effect will be weakened. Considering the control effect and engineering economy, it is suggested that the ground improvement depth $h = 2H_e$.

(4) The control effect on tunnel displacement can be increased by increasing the ground improvement width. In actual engineering, it is suggested to increase the ground improvement width as much as possible if there is enough site space to allow ground improvement and ensure the safety of the tunnel.

(5) After ground improvement using the suggested parameters, the horizontal and vertical displacements of the tunnel are significantly reduced, and the influenced zone corresponding to the same tunnel displacement isoline is reduced. According to different tunnel displacement control standards, the range outside of the pit is divided into: II–secondary influenced zone, III—general influenced zone, IV—weak influenced zone, and the I—primary influenced zone no longer exists.

**Author Contributions:** Conceptualization, B.L.; methodology, B.L.; software, B.L.; validation, B.L., and W.X.; formal analysis, B.L.; investigation, B.L.; resources, C.S.; data curation, B.L.; writing—original draft preparation, B.L.; writing—review and editing, C.S.; visualization, W.X.; supervision, C.S.; project administration, C.S.; funding acquisition, B.L. All authors have read and agreed to the published version of the manuscript.

**Funding:** This research was funded by the Natural Science Foundation of Jiangsu Province (Grant No. BK20220856), the China Postdoctoral Science Foundation (Grant No. 2021M690624), and the Jiangsu Planned Projects for Postdoctoral Research Funds (Grant No. 2021K146B).

**Institutional Review Board Statement:** Not applicable.

**Informed Consent Statement:** Not applicable.

**Data Availability Statement:** The data presented in this study are available from the corresponding author upon request.

**Acknowledgments:** Ningning Wang from JSTI Group Co., Ltd., Nanjing 210017, China.

**Conflicts of Interest:** The authors declare no conflict of interest.

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
