# Peer review of "Influenced Zone of Deep Excavation on Adjacent Tunnel Displacement and Control Effect of Ground Improvement in Soft Soil"

_applsci, doi:10.3390/app12189047_

Round 1

Reviewer 1 Report

Minor comments:

- The symbol used to represent the maximum horizontal displacement of the retaining wall in line 178 and in the title of the abscissa axis in Figure 6 (a) are different.

- Figures 7 (a) and (b) show the term DW for retaining wall. It is not clear whether it is that or it means something else, as DW is not defined in the article.

Author Response

Point 1: The symbol used to represent the maximum horizontal displacement of the retaining wall in line 178 and in the title of the abscissa axis in Figure 6 (a) are different.

Response 1: Thanks for your comments. The abscissa axis in original Figure 6 (a) has been modified to δhm, which is consistent to that in line 178.

Point 2: Figures 7 (a) and (b) show the term DW for retaining wall. It is not clear whether it is that or it means something else, as DW is not defined in the article.

Response 2: Thanks for your comments. DW is the abbreviation of diaphragm wall, and it has been clarified in the revised manuscript.

Reviewer 2 Report

The paper presented the overview about influenced zone of deep excavation on adjacent tunnel displacement and evaluation of control effect of ground improvement in soft soil. The contend of the paper is so interesting. The reviewer believes that this paper is great attention from practical reader and researcher. However, the authors should be carefully explained some works in the paper:

1.    The abstract and introduction must to emphasize the novelty of this research

2.    Please show the limitation of this study

3.    The important work is numerical studied cases 2.1.3, the author must summarize all investigated case in a table.

4.    Please update more references for using HSS model and influence of deep excavation on adjacent structure

https://doi.org/10.1007/s41062-021-00621-x

https://doi.org/10.1007/s40515-020-00142-7

Author Response

Point 1: The abstract and introduction must to emphasize the novelty of this research.

 Response 1: Thanks for your suggestion. In this research, the control effect of ground improvement on the influenced zone of deep excavation is evaluated, and the optimal parameters of ground improvement are given, which is the novelty of this research and has been supplemented in Abstract.

Point 2: Please show the limitation of this study.

Response 2: Thanks for your suggestion. The influenced zone of deep excavation on adjacent tunnel displacement and the control effect of ground improvement are influenced by many factors, such as geological conditions, structure forms, construction workmanships. In this study, the braced deep excavation with a depth of 18 m and shield tunnel with a diameter of 6 m in thick soft soil are shoosen for study, and some preliminary results are obtained. For deep excavations and tunnels under the others conditions, the results need to be further discussed and verified. The limitation has been supplemented in Discussion.

Point 3: The important work is numerical studied cases 2.1.3, the author must summarize all investigated case in a table.

Response 3: Thanks for your suggestion. The table 2 has been supplemented to summarise the numerical simulation conditions.

Point 4: Please update more references for using HSS model and influence of deep excavation on adjacent structure, https://doi.org/10.1007/s41062-021-00621-x, https://doi.org/10.1007/s40515-020-00142-7

Response 4: Thanks for your suggestion. The above references has been supplemented in the revised manuscript.
